# OpenReview forum: "Semi-supervised Long-tailed Recognition using Alternate Sampling"
_ICLR.cc/2024/Conference — Submitted to ICLR 2024_

### Official Review · Reviewer_9E5Z · 2023-10-15

**Soundness:** 2 fair
**Presentation:** 2 fair
**Contribution:** 1 poor
**Rating:** 3
**Confidence:** 5

**Summary:**

This study aims to train an algorithm with a training set that class distribution is imbalanced and it has both labeled and unlabeled sets. For the training, the authors borrowed the concept of decoupled learning of features and classifier. Specifically, the authors proposed an algorithm consists of three stage which separatively and iteratively learns features and classifier using alternative sampling and pseudo-labels. Experiments on two datasets verify effectiveness of the proposed algorithm.

**Strengths:**

Many SSL studies and long-tailed recognition studies are reviewed in Section 2.

Introduction is easy to read.

**Weaknesses:**

Code is not submitted. Thus I cant reproduce the experimental results.

There is no theoretical ground for the proposed algorithm.

The proposed algorithm is outdated. Specifically, there are already many long tailed semi supervised learning (LTSSL) algorithms such as DARP, ABC, DASO, SAW, CReST, CoSSL, Adsh, DebiasPL, L2AC and UDAL. None of them were reviewed and compared in the experiments section. I think the algorithm and draft should be significantly revised.

Whereas other LTSSL algorithms conducted experiments on CIFAR-100-LT, STL-10-LT, the proposed algorithm did not conduct experiments on the datasets.

Experimental settings are not same with that of the other LTSSL studies.

**Questions:**

I have no question.

---

### Official Review · Reviewer_a8kj · 2023-10-26

**Soundness:** 2 fair
**Presentation:** 3 good
**Contribution:** 2 fair
**Rating:** 3
**Confidence:** 4

**Summary:**

This paper studies the semi-supervised long-tailed recognition problem, which aims to leverage unlabeled data to enhance performance on long-tailed recognition. An alternate sampling method is proposed to incorporate the decoupling in long-tailed recognition with semi-supervised learning. The improved performance is reported in the paper.

**Strengths:**

1.	The setting is interesting when considering long-tailed learning and semi-supervised learning.
2.	This paper is easy to follow.

**Weaknesses:**

1.	A similar setting has been discussed in some papers [1, 2, 3], and I cannot find any difference with the setting proposed here. So I advise authors to compare the performance with the mentioned methods or discuss the difference between these two settings.
2.	The proposed method seems to be a combination of decoupling and semi-supervised methods. The novelty should be explained more clearly.
3.	The experiments conducted are not enough to demonstrate the effectiveness of the method. The compared methods are too weak in this setting.
4.	From the experiments, the performance for the few-shot classes is not strong enough.

[1] Wei T, Gan K. Towards Realistic Long-Tailed Semi-Supervised Learning: Consistency Is All You Need[C]//Proceedings of the IEEE/CVF Conference on Computer Vision and Pattern Recognition. 2023: 3469-3478.
[2] Lee H, Shin S, Kim H. Abc: Auxiliary balanced classifier for class-imbalanced semi-supervised learning[J]. Advances in Neural Information Processing Systems, 2021, 34: 7082-7094.
[3] Oh Y, Kim D J, Kweon I S. Daso: Distribution-aware semantics-oriented pseudo-label for imbalanced semi-supervised learning[C]//Proceedings of the IEEE/CVF Conference on Computer Vision and Pattern Recognition. 2022: 9786-9796.

**Questions:**

see weakness

---

### Official Review · Reviewer_TaGn · 2023-10-30

**Soundness:** 2 fair
**Presentation:** 2 fair
**Contribution:** 2 fair
**Rating:** 5
**Confidence:** 4

**Summary:**

In this paper, the authors address long-tailed recognition in the semi-supervised learning paradigm. The authors claim two long-tailed recognition challenges: sample scarcity and imbalanced class distribution, which is solved by the available unlabeled data. This paper proposes an alternate sampling strategy and consistency regularization in a decoupling framework, i.e., iteratively optimizing representation learning and classifier learning. Experiments are conducted on CIFAR-LT and ImageNet-LT to validate the proposed method.

**Strengths:**

- Leveraging unlabeled data is reasonable to mitigate the negative impact of long-tailed data distribution.
- Generally, the paper is easy to follow.

**Weaknesses:**

- The related work part is outdated. Many imbalanced semi-supervised learning methods are not discussed and compared.
- The technical contribution is not clear. The consistency loss is similar to an early study in the semi-supervised learning [1], alternating sampling is also straightforward in the long-tailed learning methods.
- Lack of comprehensive evaluation (e.g. imbalanced ratio, unlabeled data ratio…) and ablation studies (hyper-parameters, different sampling strategy…).
- The experiments are not adequate. The authors should conduct experiments on representative long-tailed benchmark (Places-LT and iNaturalist). The chosen supervised long-tailed baselines are not state-of-the-art methods. The authors should conduct experiments compared with more recent methods (Logit adjustment[2], parametric contrastive learning[3]...). Besides, the authors should compare with semi-supervised long-tailed methods.

[1] Laine et al., "Temporal Ensembling for Semi-Supervised Learning. "

[2] Menon et al. "Long-tail learning via logit adjustment. "

[3] Cui et al. "Parametric contrastive learning."

**Questions:**

Please refer to the weakness part.

**Details Of Ethics Concerns:**

No.

---

### Official Review · Reviewer_3b2a · 2023-11-01

**Soundness:** 3 good
**Presentation:** 3 good
**Contribution:** 1 poor
**Rating:** 3
**Confidence:** 4

**Summary:**

The semi-supervised long-tailed recognition problem is defined in this article, which also combines the long-tailed and semi-supervised problems in recognition tasks. To present a three-stage learning approach, the author integrates certain techniques from the previous long-tailed learning field and the semi-supervised learning field. Before the three stages of learning, the author, who refers to the previous methods, decouples feature embedding and classifier to initialize the model. Afterward, the model undergoes several rounds of learning, repeating the methods of these three stages in each round. In the first stage, the model generates pseudo labels on unlabeled data. In the second stage, the model uses both pseudo-label samples and labeled samples for training feature embedding. In the third stage, the model uses labeled samples to train the classifier. Among them, random sampling is used in the first and second stages, and class-balanced sampling is used in the third stage. This method is an extension of the decoupling method in the field of long-tailed learning on semi-supervised data, which helps to make model representation more generalized and classification more balanced. The author also demonstrated in the experiment the superior performance of such methods on extremely imbalanced data.

**Strengths:**

S1. This article combines the fields of long-tailed learning and semi-supervised learning, opening up a new field. By combining methods from both fields, there is a certain improvement in model performance,  and the improvement effect is very significant in extremely unbalanced scenarios.

**Weaknesses:**

W1. The method proposed in this article only performs outstandingly on extremely imbalanced datasets, with average performance improvement on datasets with low imbalance.

W2. In addition, this method heavily relies on the initial accuracy of the model on the dataset. If the initial accuracy is high and the generated pseudo labels are more accurate, the performance improvement will be significant. When the initial accuracy is low, the performance improvement is very limited, so the effectiveness of this method is relatively limited.

**Questions:**

Q1. A similar article (https://arxiv.org/abs/2105.00133) was published on Arxiv in 2021, and there have been other follow-up studies. Your article states that this is the first article to propose semi-supervised long-tailed learning. Is it slightly inappropriate to place this at this time point in 2023?

Q2. Can you explain the relationship between this article and the article published on Arxiv in 2021? If it is a follow-up study, can you introduce relevant improvements?

**Details Of Ethics Concerns:**

Similar articles (https://arxiv.org/abs/2105.00133) were Published on Arxiv in 2021, with a similarity of 90%. Perhaps it is necessary to check if the authors of both are the same.

---

### Meta-Review · Area_Chair_2oyP · 2023-12-05

**Metareview:**

This paper studies semi-supervised long-tailed recognition problem, where the data exhibits a long-tailed imbalanced distribution, and there is unlabeled data that can be leveraged. To address this problem, the authors incorporate the idea of model decoupling and balanced sampling. Furthermore, it proposes a three-stage approach to learn from both labeled and unlabeled data. The empirical results demonstrate the effectiveness of the proposed method.
The studied problem is interesting and a corresponding method is proposed. However, all of the reviewers are with negative comments. The proposed approach is a easy combination of previous methods. Moreover, there are already many long-tailed semi-supervised learning approaches, but the authors have not discussed and compared them. More concerns can be referred to from the reviewers' comments, such as unsatisfactory experiments, code reproducibility, and no theoretical ground.
Moreover, the authors have not made any response to the reviewer' comments, and the concerns are not addressed at all. Therefore, I recommend rejecting this paper.

**Justification For Why Not Higher Score:**

This paper studies an interesting and popular problem, which has been studied by several works before. However, this paper has not discussed the previous works. Moreover, the proposed method has limited novelty, and empirical study is trivial. Therefore, I think this paper has limited contribution and recommend the rejection.

**Justification For Why Not Lower Score:**

N/A

---

### Decision · Program_Chairs · 2024-01-16

Reject